# Virtual Surgical Reduction in Atrophic Edentulous Mandible Fractures: A Novel Approach Based on "in House" Digital Work-Flow

**Vincenzo Abbate** [1,*], **Umberto Committeri** [1], **Stefania Troise** [1], **Paola Bonavolontà** [1], **Luigi Angelo Vaira** [2], **Guido Gabriele** [3], **Federico Biglioli** [4], **Filippo Tarabbia** [4], **Luigi Califano** [1] and **Giovanni Dell'Aversana Orabona** [1]

[1] Maxillofacial Surgery Unit, Department of Neurosciences, Reproductive and Odontostomatological Sciences, University of Naples Federico II, Via Sergio Pansini 5, 80131 Naples, Italy
[2] Maxillofacial Surgery Operative Unit, University Hospital of Sassari, Viale San Pietro 43/B, 07100 Sassari, Italy
[3] Maxillofacial Surgery Operative Unit, University Hospital of Siena, Via Banchi di Sotto 55, 53100 Siena, Italy
[4] Maxillofacial Surgery Operative Unit, ASST Santi Paolo e Carlo, Via Antonio di Rudinì, 8, 20142 Milano, Italy
* Correspondence: vincenzo.abbate@unina.it; Tel.: +39-3357-266-177

**Featured Application: Virtual Surgical Planning in Cranio-Facial Traumatology.**

**Abstract:** Atrophic edentulous mandible fractures are a challenge for maxillo-facial surgeons because of low vascularization, low bone regeneration, and lack of occlusion. Whereas occlusion is the main guide in the reduction of mandibular fractures, the aim of our study is to show the advantages of using virtual surgical planning (VSP) in surgery when the occlusal guide is absent. This work is a prospective study that shows the in-house digital workflow for the management of these fractures in the Maxillo-Facial Surgery Unit of Federico II University Hospital of Naples. Four patients who satisfied the criteria were included in the study. For each patient, the same defined CAD/CAM-based was applied. The workflow followed four steps: (1) bone segmentation and virtual reduction of fracture fragments; (2) three-dimensional printing of virtually reduced mandible and modelling of 2.4 reconstruction plate on printed resin model; (3) surgery aided by the pre-formed plate; (4) digital and clinical outcomes analysis. In the last step, a distance colour map was conducted to compare the virtual planning and postoperative CT outcome. In all cases, the discrepancies values between the two images were lower than 1.5 mm, and good clinical outcomes in terms of facial symmetry, absence of sensory disturbance, and possibility of prosthetic rehabilitation were obtained. In conclusion, the VSP, with our in-house workflow brings benefits in the management of atrophic edentulous mandible fractures in terms of the high accuracy of bone repositioning.

**Keywords:** atrophic edentulous mandible; mandibular fractures; virtual surgical planning; reconstruction-plate; digital workflow; CAD/CAM technology

## 1. Introduction

The incidence of fractures in atrophic edentulous mandibles has been reported to range between 1% to 5% of all mandibular traumas; in elderly people, these fractures range between 10.1% to 56% due to poor proprioception, weakness, and impaired reflexes, which lead to a greater frequency of accidental falls. The indications to treat these fractures are mainly based on the possibility to guarantee a good quality of life in terms of chewing, swallowing, phonation, displacing of bone stumps, and the possibility of prosthetic rehabilitation [1]. Therefore, the goal of surgical treatment is to restore anatomical continuity and facial symmetry to obtain valuable prosthetic rehabilitation. Open Reduction Internal Fixation (ORIF) represents the most valuable approach [2–5], but the management of such kinds of fractures is still a challenge for the technical difficulties and the comorbidities that may afflict elderly patients.

In such cases, the fragments' anatomical reduction and fracture consolidation are difficult to achieve due to bone atrophy which lowers bone regeneration and causes poor vascularization regularly guaranteed by the periosteum. Thus, a load-bearing plate is mandatory due to the decrease in bony volume and vascularization in these high-risk patients. However, the lack of occlusions is the main problem in these fractures since occlusions are the main guide for a stable reduction. Surgeons face difficulties in operating with no references to align the bone stumps [6,7] in patients who lack dental elements. To solve these difficulties, computer-assisted design (CAD) and computer-assisted manufacturing (CAM) technology are increasingly used in clinical practice. The ability to virtually program surgical procedures and transfer the planning to the operating room is now finding more and more applications in maxillofacial surgery [8,9]. The main application fields of VSP in maxillofacial surgery were orthognathic/malformation surgery, implant surgery, and oncological-reconstructive surgery. The introduction of VSP into craniofacial traumatology is certainly more recent and it has been mainly documented for complex fractures, especially for comminuted fractures, and in cases with fragments displacement. The efficacy of the analogical procedure has been confirmed for orbital, zygomatic, and dentate mandible fractures, but there is not sufficient evidence regarding fractures of edentulous atrophic mandibles. Based on the previous evidence, CAD/CAM (or VSP) technologies could be a support to the surgeon in the treatment of these complex fractures. Hence, the scope of this paper is to present a new in-house digital workflow aimed to finalize a patient-specific implant (PSI) as a reference for effective Open Reduction and Internal Fixation (ORIF) when the occlusal guide is absent.

## 2. Materials and Methods

This work is a prospective study that shows the in-house digital workflow for the management of complex fractures in atrophic edentulous mandibles. The study was conducted in the period between October 2020 and March 2021 on patients admitted to the Maxillo-Facial Surgery Unit of Federico II University Hospital in Naples. The study met the criteria established by the Declaration of Helsinki and was approved by the Ethics Committee of the Federico II University hospital of Naples with protocol number 373/19. The patients enrolled in the study satisfied the following inclusion criteria:

- post-traumatic comminuted mandibular fractures verified with a CT scan within 12 h from the trauma
- atrophic edentulous mandible
- unavailability of the patient's personal dental prostheses that could guide the occlusion; written informed consent to undergo the surgical procedure

Patients who met the following criteria were excluded:

- previous surgically treated mandibular fractures
- patients who have refused surgical treatment
- partial edentulism
- taking drugs that affect bone resorption

Among nine patients likely to be enrolled, only four patients satisfied the inclusion criteria and were enrolled in the study; two patients presented partial edentulism but enough to allow a fracture reduction based on the occlusion; one patient had tumour pathologies and took bisphosphonates medications, causing a poor bone quality; one patient refused surgical treatment; one patient previously underwent surgical treatment for mandibular neoplasia.

Among the four patients enrolled in the study, a computer-assisted digital workflow was settled to perform the virtual reduction of the fractures and to finalize a patient specific plate (PSP) to fix the mandible. For each patient, the same CAD/CAM based workflow was applied as defined below:

(1) Digital Imaging and Communications in Medicine (DICOM) to Stereolythography (STL) files: by using Materialize Mimics Medical 21.0 software (Technologielaan 15,

3001 Leuven, Belgium) to perform the virtual reduction. The first step was to upload the DICOM-Files to generate a 3D model of the patient. Then, the segmentation of bone fragments was possible thanks to the New Mask tool. Generally, the range of bone threshold was between 0–1250 HU. Once the first step was completed, a three-dimensional image of the entire splanchnocranium was obtained. The division of each bone fragment was needed to proceed with the simulation of the fracture reduction. Therefore, in the second step, the Split Mask tool allowed to divide of the main mask into single segments for each bony fragment. In this way, each fragment could be repositioned virtually mimicking the fracture reduction, while the optional function Smart Fill was used to fill the bone cavities. The process of identification of the fracture lines was performed by three different maxillo-facial surgeons (VA, UC and ST). The produced images were then converted into "objects files". Using the 3-Matic Medical 3D (Materialise, Leuven, BE) it was possible to convert the stumps into a 3D plan and obtain a virtual reduction with the stumps in the correct anatomical position. To avoid any bias, three different maxillofacial surgeons (VA, UC and ST) also performed the procedure. The reduction was obtained by considering the alignment of the condyles in the glenoid fossa. At the end of the procedure, the virtually reduced mandible file was converted into a STL file adapted for the "Formlab-Form 3B" 3D printer (Figure 1a,b).

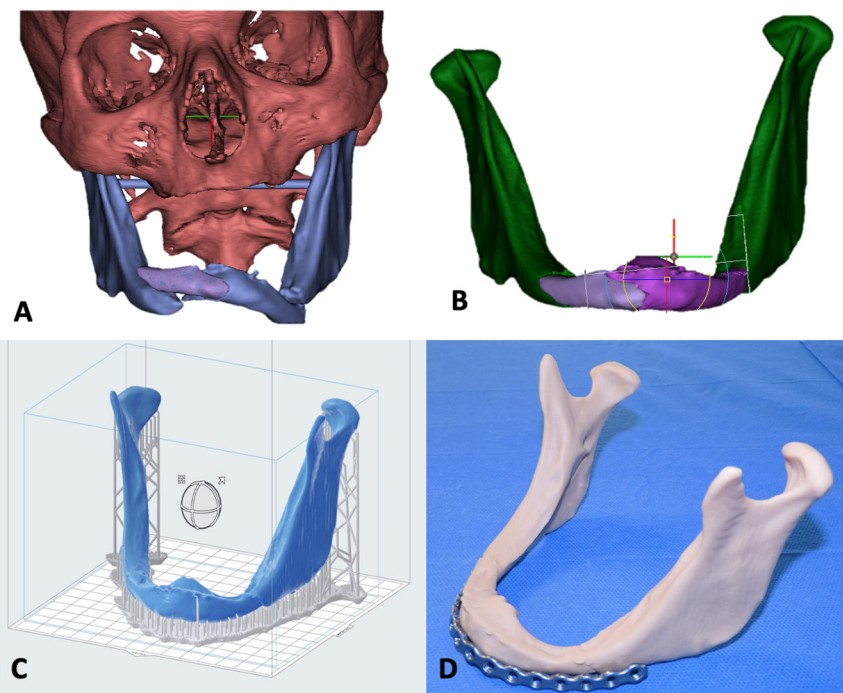

**Figure 1.** "In-house" workflow. (**A**) Segmentation of the fracture segments in the pre-operative CT using the Materialize Mimics 21.0 Medical Software; (**B**) virtual reduction of fracture fragments; (**C**) 3D printing setting of the virtual planned reduced mandible using the Preform 3D Printing Software; (**D**) Modeling of the plate on the reduced mandible resin model.

(2) In House Rapid prototyping 3D Print—Resin model and 2.4 reconstruction plate modelling: a specific slicing software (PreForm 3D Printing Software—Formlabs 3B+) was used to set the model for 3D printing. It was necessary to use the supports to have the highest possible quality of the model (Figure 1c). A resin StereoLithography Apparatus (SLA) 3D printer (Formlab-Form 3B+, located in Somerville, MA, USA) was used for in-house rapid prototyping. The selected material was the Formlab Model Resin V3 (ISO 10993-5: 2009) because of its mechanical characteristics such as tensile strength of 27 MPa and modulus of elasticity of 1.1 GPa. Once the print was completed, the model was subjected to an autonomous immersion bath in isopropyl alcohol using the Form Wash

(Form Wash -located in Somerville, MA, USA) device for 20 min. The final step was the photopolymerization using the Form Cure (Form Cure located in Somerville, MA, USA) device that uses the UV order to increase the tensile strength up to 48 MPa and modulus of elasticity up to 2.3 GPa.

Once the reduced mandible model was obtained, a 2.4 reconstruction plate was modelled to be applied during the surgery (Figure 1d). The plate was bent by applying manual pressure to finalize a customized device for each patient and was sterilized in an autoclave with a temperature of 160 °C for 60 min, 24 h before surgery.

(3) Performing surgery: all the surgeries were performed by the same surgical team and under general anesthesia. A small cutaneous submental surgical access was performed for an extension of the fracture of about 6 cm in each operation. After soft tissue detachment, the bone fragments were identified, and the cortical bone was skeletonized to avoid the complete elevation of the periosteum and not to create further loss of the vascularization. Then, the plate was positioned between the two fixed bone stumps of mandibular body. The fragments were replaced in the correct anatomic position, and the reduction of the comminuted fracture of the mandible was possible based on the previously modeled plate. At the end of the procedure, the surgeon checked the facial symmetry, joint functions, and condyles correctly in the glenoid fossa bilaterally. A nylon 4.0 was used for the final suture. A surgical procedure example is shown in Figure 2a–d.

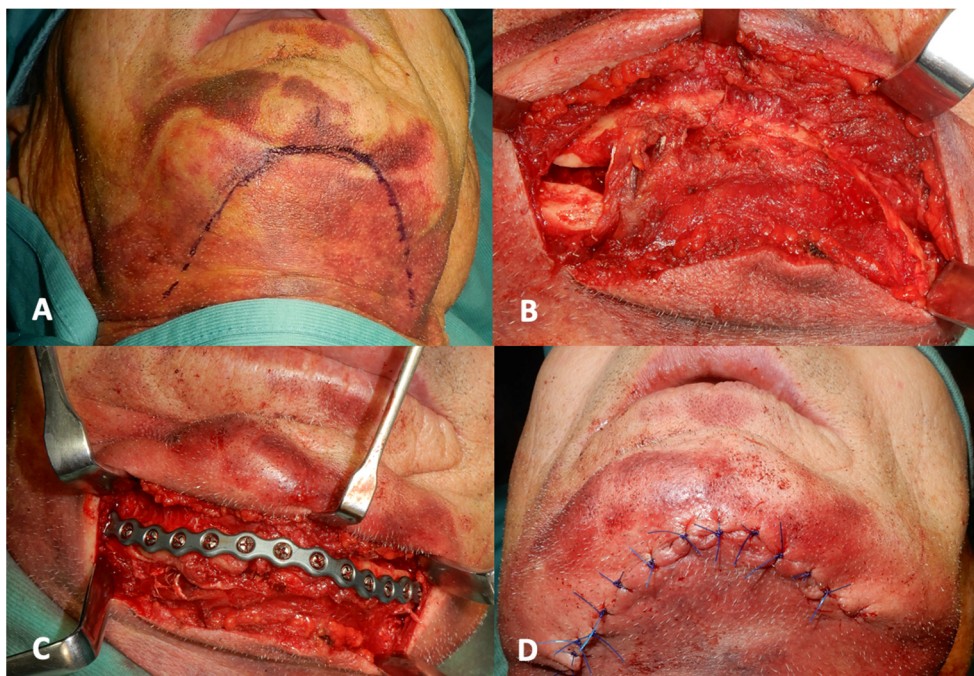

**Figure 2.** Surgery. (**A**): submental approach; (**B**): exposure of the fracture fragments; (**C**): ORIF of fracture with preformed plate; (**D**): suture in nylon.

(4) Initial digital outcomes: all the patients underwent a post-surgical CT scan the following day of the surgery. By using Materialize Mimics Medical 21.0 software, DICOM data have been converted into STL files. The workflow to obtain segmented mandible was the same described previously. The Geomagic Design X software (3D point cloud and mesh processing software, 11 Breedewues, L-1259 Senningerberg—Production, Logistics & Service Center) was used to compare the planned STL file in a 3D space obtained by our digital planning procedure with the post-surgical STL file. A distanced color map was obtained overlapping the two images and the discrepancies in mm between these two images in all four cases (Figure 3). To reduce any human errors in the overlapping process, the non-manual but automated protocol of the Geomagic software was used. The upper/lower limit for color coding of the discrepancies was fixed as +2 mm and

−2 mm, so that deviations appeared in different colors. Steps of 0.25 mm were used, so each color encoded a distance interval of 0.25 mm. A total discrepancy at the cloud point was calculated between the two images. However, to avoid the bias of the plate in the postoperative image the discrepancies were also calculated manually at the level of well defined bone points. Therefore, 11 anatomical bone landmarks were set to compare the "planned model" and the "in vivo" result on the following cephalometric points: menton, pogonion, B point, left and right condyle, left and right gonion, left and right lingula, left and right mental foramen.

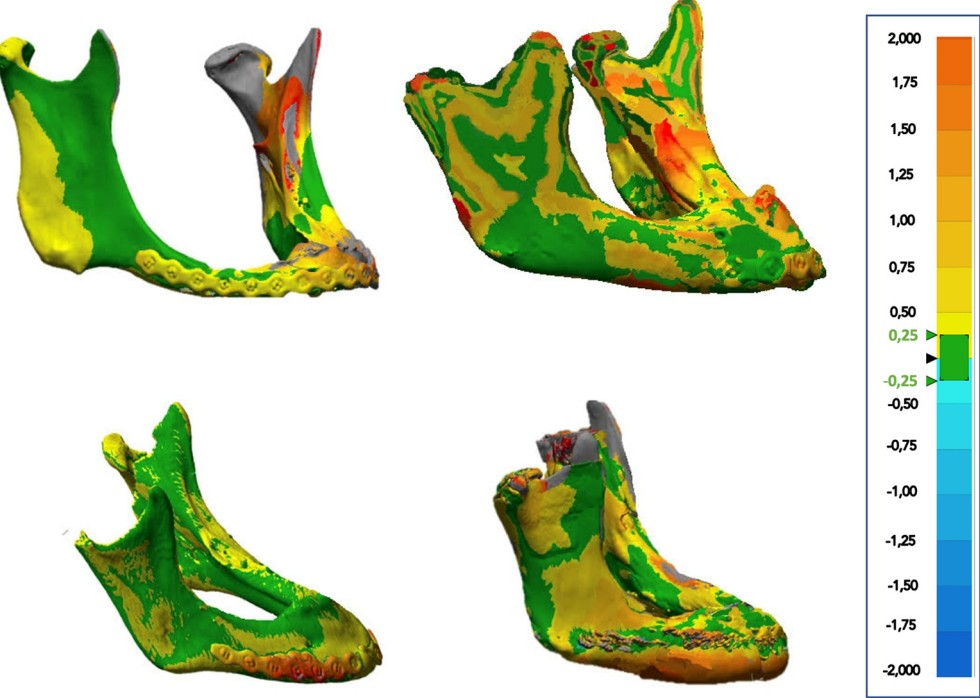

**Figure 3.** Digital analysis. Distance color map of the four cases. On the right the color legend bar.

(5) Long-term outcomes: all the patients were clinically evaluated during the outpatient follow-up. The patients went through periodic controls at 1 month and at six months period. During the outpatient controls aimed to check/control the facial symmetry, pain, joint and masticatory function, mouth opening, sensory, and motor alterations related to deficits of the V and VII cranial nerve.

## 3. Results

Four patients were treated using the CAD/CAM and 3D printing technology as the described protocol. The sex was equally distributed, the average age was 79 years (range 61–88 years). The operating times were very short, with an average surgical time of 45 min. No complications were observed during surgery or at the immediate post-surgical care. The average hospitalization time after surgery was four days (ranging from two to six days). At the six-monthly follow-up, we observed good clinical outcomes in our samples in terms of facial symmetry restoration, absence of sensory disturbances of the inferior alveolar nerve, no facial nerve weakness, satisfying joint and masticatory function, and mean mouth opening of 37 mm (rangin from 33 to 40 mm).

Regarding the digital workflow, the virtual planning design took around 2 h for each case while about 6 h for each case for the printing process and sterilization. On average, about $45 \pm 2$ mL of resin was required to print the models. An estimated cost of about 6.3 Euros is estimated for each case. The discrepancy values deriving from the overlapping analysis of the models were shown in Table 1 (Table 1). In the examined samples, all the obtained discrepancies' values were less than 1.5 mm. The mean value of the discrepancies

of the whole overlapping cloud point was 0.69 ± 0.33 mm. Regarding the discrepancies of the cephalometric examined points in the sample, the higher mean value was obtained in the condylar site (Right: 0.80 ± 0.26 mm—Left: 0.79 ± 0.32 mm), while the lower mean value was obtained in the B Point (0.55 ± 0.41 mm).

**Table 1.** Discrepancies analysis.

| | CASE 1 | CASE 2 | CASE 3 | CASE 4 | Average ± SD |
|---|---|---|---|---|---|
| | Discrepancy mm | Discrepancy mm | Discrepancy mm | Discrepancy mm | Discrepancy mm |
| Whole cloud point | 0.63 | 0.48 | 0.72 | 0.94 | 0.69 ± 0.33 |
| Menton | 0.46 | 0.24 | 0.72 | 0.88 | 0.57 ± 0.49 |
| Pogonion | 0.48 | 0.32 | 0.67 | 0.84 | 0.58 ± 0.34 |
| B Point | 0.44 | 0.28 | 0.69 | 0.81 | 0.55 ± 0.41 |
| Right Condyle | 0.72 | 0.67 | 0.81 | 1.01 | 0.80 ± 0.26 |
| Left Condyle | 0.71 | 0.63 | 0.79 | 1.05 | 0.79 ± 0.32 |
| Right Gonion | 0.55 | 0.46 | 0.57 | 0.92 | 0.62 ± 0.35 |
| Left Gonion | 0.58 | 0.44 | 0.54 | 0.89 | 0.61 ± 0.34 |
| Right Mental Foramen | 0.67 | 0.35 | 0.61 | 0.82 | 0.61 ± 0.34 |
| Left Mental Foramen | 0.64 | 0.33 | 0.63 | 0.79 | 0.60 ± 0.33 |
| Right Lingula | 0.66 | 0.49 | 0.66 | 0.94 | 0.69 ± 0.32 |
| Left Lingula | 0.65 | 0.41 | 0.68 | 0.97 | 0.68 ± 0.40 |

## 4. Discussion

The management of atrophic jaw fractures has always been a topic of discussion in the literature. Luhr et al. [10] in 1996 already classified the atrophic mandibles into three categories, according to the degree of atrophy and the mandibular height: Class I, 16 to 20 mm; Class II, 11 to 15 mm; Class III, <10 mm. In the Class III, bone quality diminished because of a possible sclerotic and because of blood flow decrease. After Luhr et al.'s studies, several authors focused on the correct management and treatment of Class III fractures that often are atrophic pluri-fragmentary mandibular fractures. The problems related to these fractures are associated with patient's age, medical comorbidities, poor bone quality, and decreased vascularity, as well as reduced contact area between the fracture ends [7]. In addition, mandibular atrophy is often the result of complete edentulism: this condition determines the loss of the occlusal reference, which is the guide for the correct reduction of these fractures [6,7,11]. In the literature, the possible treatments for atrophic edentulous mandibular fractures are:

(1) Observation: in case of not dislocated fractures in patients with severe anesthetic risk. The most frequent complication is the non-union of the bone stumps [1];

(2) Closed reduction (mandibulomaxillary fixation—MMF): in case of edentulous atrophic fractures in patients with high anesthetic risk; postoperative malunions and nonunions were very frequent [12];

(3) Gunning splints: the use of gunning splints is preferred in case of edentulous atrophic patients because the open reduction is not helpful due to compromised medical condition of these patients. However, as Dharaskar et al. have shown, there is the possibility of ankylosis, induced by joint blockage (5–6%) [13];

(4) External fixation: it is indicated as a temporary fixation when the patient needs earlier medical treatment. External fixators do not guarantee permanent stability, so malunion and nonunion are common [14];

(5) Open Reduction Internal Fixation (ORIF): with titanium mesh, locking miniplate, or 2.4-reconstruction plate, with or without simultaneous bone grafting. It is indicated for all atrophic surgical mandible fractures. In all cases, the aim of the treatment of edentulous

atrophic mandibular fracture should be to improve patient's quality of life with the minimal risk, ensuring fracture stability [15,16].

Bruce and Ellis (1993) in their review concluded that the optimal treatment for this kind of fractures is an open reduction and stable fixation with large osteosynthesis plates (ORIF) [17]. Different types of techniques were used for this goal such as fixing the bone stumps in occlusion with the patient's dentures [18] or fixing miniplates to the inferior mandibular border to temporarily maintain the fragments alignment before applying a reconstruction plate to vestibular cortical bone [19]. Considering that the use of mini plates can increase the operating times and, furthermore, the patient's dentures are not always available, using a pre-formed plate can become a useful resource both for guiding the reduction in absence of occlusal guide and for reducing intraoperative times. In order to pre-model a 2.4 reconstruction plate, we used Virtual Surgical Planning (VSP). The recent scientific literature has revealed that VSP is a valid tool of assisting surgeons in various procedures, ranging from orthognathic surgery to reconstructive surgery with free flaps, correction of congenital malformations and craniosynostosis, cranio-facial traumatology, distraction osteogenesis, and implantology. Although the use of VSP has been reported in all cited fields, currently there is minimal evidence in elderly patients with atrophic edentulous mandibular fractures [20,21]. In mentioned studies, the authors underlined the importance of VSP to solve the problems related to intraoperative reduction of fragments and to postoperative complications. In fact, without the use of VSP, different complications have been reported: pseudarthrosis of the bone stumps, osteomyelitis, unstable reduction of the fragments that did not allow prosthetic rehabilitation, facial asymmetry, and sensory disturbances of the inferior alveolar nerve [22]. Moreover, the majority of the works in the literature pave the way for the VSP treatment through case reports or case series, but there are no randomized controlled trials on a large sample. [23,24] For this reason, new and extensive contributions to the topic are supported to increase the collective experience in this field. Thus, the aim of our study has been to provide our experience of an in-house digital workflow to show the advantages of VSP in the management and treatment of atrophic edentulous mandibular fractures, where the occlusal guide is absent.

The applied protocol aims to model a reconstruction plate based on the patient's anatomy. Indeed, the time spent for the design phase guarantees a reduction of the operating times. The accuracy of the printing depends on the accuracy of the CT, which must have slices of 1–2 mm. The Formlabs 3B+ 3D print allows to create details with microscopic precision (resolution 25 micrometres); this is a fundamental characteristic for printing instruments and devices in the medical and surgical fields. The thickness of each layer can vary between 25–300 micrometres, for high precision of the anatomical details of the model.

Regarding the surgical procedure, Brucoli et al. [7] in their paper confirmed that, for the treatment of fractures in Luhr's Class III mandibles, the cutaneous external surgical approach is preferred. In fact, this access ensures adequate fragments exposure and, consequently, a more careful surgical procedure. It reduces the risk of contamination by oral microbes and prevents any injury to the inferior alveolar nerve, usually located on top of the alveolar crest in atrophic mandibles [25]. In accordance with the literature, we applied this approach in all patients: thanks to the pre-formed plate, we avoided a cervicotomy to expose the bone fragments, but we used a small sub-mental access. In fact, starting from a minimal sub-mental access, we exposed all the fractures stumps moving the retractors according to the surgical phase.

Recent studies [26] have showed that the use of a pre-formed reconstruction plate can reduce both the time of the manual plate modeling during the surgery and the time for finding the correct position of the plate on the mandibular fragments.

In our cases, we firstly positioned the pre-formed plate on the resin model to calculate the correct position and to estimate the correct screws' number to insert on each mandibular stump. During the surgical procedure, we found that the plate fits perfectly on the bone surfaces, and no further manipulation of the plate was required. Finally, the other fragments

were reduced and fixed to the plate as the pre-operative program. At the end of procedure, the surgical time was significantly reduced compared to our standard time, and the accuracy of the treatment was greatly improved.

The success of the treatment was confirmed by the comparison between the virtual surgical planned CT and the post-operative CT. In fact, the overlapping of the two CT images showed that the discrepancies in mm were in all cases less than 1,5 mm. In particular, to verify where the outcome was more predictable, we calculated the discrepancies in eleven bone cephalometric points. The analysis showed that the front portion of mandible (B point and Menton) presented the lower discrepancy, while the higher discrepancy was obtained in the posterior section (condyles and Lingula). This result seems to confirm that the position in the space of the anterior fragments is more easily predictable than the posterior sections. This is probably due to the action/force of the applied plate to the bone that stabilizes the bony portion compared to other areas which are more unstable.

These results were confirmed by the good clinical outcomes in all four cases in terms of facial symmetry restoration, absence of sensory disturbance and possibility of a prosthetic rehabilitation. Furthermore, it is a very low-cost method. If we exclude the initial investment for the purchase of the printer and the dedicated software (around 15,000.00 Euros), the cost for single case is estimated to be around 6.3 Euros. A patient specific plate (PSP) has a commercial cost of around 5000.00 Euros. Therefore, the described in-house workflow allows to amortize initial set-up expenses after a few performed cases.

One of the limitations of the study is certainly the sample size. This is a small sample enrolled in a single center. Further studies or larger case studies with the involvement of other structures are necessary to confirm the results obtained. Other limits are represented by the long learning curve and the time consuming. In fact, the procedure requires specific staff with a background in CAD CAM technologies. This method is reproducible in other clinics by having the medical software and the three-dimensional printer. Another limit of the study is the possible reduction in the plate resistance compared with a custom-made plate, which must not be modeled. However, this reduction of resistance is lower than the conventional method because the plate is subject to less stress, given that it is modeled on a template.

We suggest that future studies could focus on this protocol for other types of fractures, such as comminuted maxillo-mandibular fractures.

### 5. Conclusions

Within the limits of the study, the Virtual Surgical Planning could be a helpful instrument in the management of the complex fractures in atrophic edentulous mandibles both to reduce the fracture and to decrease the surgical times. The bone repositioning accuracy of our in-house digital workflow is high because all the obtained discrepancy values have been less than 1.5 mm. Nevertheless, the virtual reduction of atrophic mandible fractures is more predictable in the anterior portion.

**Author Contributions:** Conceptualization, V.A. and U.C.; methodology, S.T.; software, U.C.; validation, G.G., G.D.O. and P.B.; formal analysis, S.T.; investigation, V.A.; resources, P.B.; data curation, F.B. and G.D.O.; writing—original draft preparation, S.T.; writing—review and editing, V.A.; visualization, F.T. and L.A.V.; supervision, L.C.; project administration, L.C.; funding acquisition, L.A.V. All authors have read and agreed to the published version of the manuscript.

**Funding:** This research received no external funding.

**Institutional Review Board Statement:** The study was conducted in accordance with the Declaration of Helsinki and was approved by the Ethics Committee of University of Naples Federico II with the protocol number 373/19.

**Informed Consent Statement:** Informed consent was obtained from all subjects involved in the study. Written informed consent has been obtained from the patients to publish this paper.

**Data Availability Statement:** Not applicable.

**Acknowledgments:** Materialize Mimics 21.0 Software.

**Conflicts of Interest:** The authors declare no conflict of interest.

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
