# Peer review of "Virtual Surgical Reduction in Atrophic Edentulous Mandible Fractures: A Novel Approach Based on “in House” Digital Work-Flow"

_applsci, doi:10.3390/app13031474_

Round 1

Reviewer 1 Report

I would suggest making the text more scientific. It still sounds a bit like a sales text. Please just present facts clearly, without using lurid adjectives or filler words. Also, phrases like "nowadays" do not fit into a scientific text.

Perhaps another name can be found for "in-house" workflow.

Figure 3 is very blurred.

The whole text needs to be shortened and sharpened. The question that arises for me is whether the system can be transferred to other clinics at all? All the time, even in the title, there is talk of "in-house". So it only works in Naples? What interest would anyone else have in it?
Is this just a narrative report about how it is done in Naples?
What is the scientific question or the as yet unsolved problem behind it anyway? That is not mentioned anywhere.
So: for a scientific publication you need a description of a problem that has not yet been solved. Then comes the description of the solution. Then the strengths and weaknesses of the proposed solution. And then the suggestion of how this can be transferred to other clinics.

Author Response

Dear Sir/Madam and Colleagues, thank you very much for your suggestions and queries regarding our paper. It can be remarkably improved.

  • I would suggest making the text more scientific. It still sounds a bit like a sales text. Please just present facts clearly, without using lurid adjectives or filler words.

Thanks for the suggestion, we have reviewed the text with a native speaker’s help and we have eliminated superfluous and redundant words, making the manuscript more scientific.

  • Also, phrases like "nowadays" do not fit into a scientific text.

Thanks for the suggestion, we have provided to eliminate this word.

  • Perhaps another name can be found for "in-house" workflow.

Thanks for the comment, the term “in-house” has already been used in international literature (Database Pubmed, Scopus, Google Scholar, etc..) to indicate digital workflows based on CAD CAM and rapid prototyping technologies performed directly in hospitals (Brucculeri L, Carpanese C, Palone M, Lombardo L. In-House 3D-Printed vs. Conventional Bracket: An In Vitro Comparative Analysis of Real and Nominal Bracket Slot Heights. Applied Sciences. 2022; 12(19):10120/ Marschall JS, Dutra V, Flint RL, Kushner GM, Alpert B, Scarfe W, Azevedo B. In-House Digital Workflow for the Management of Acute Mandible Fractures. J Oral Maxillofac Surg. 2019 Oct;77(10):2084.e1-2084.e9. doi: 10.1016/j.joms.2019.05.027.). This procedure allows to bypass external prototyping reducing costs and times that, as you know, are essential in the treatment of fractures.

  • Figure 3 is very blurred.

Thanks for the suggestion, we have provided to modify the figure 3.

  • The whole text needs to be shortened and sharpened.

Thanks for the suggestion, we have provided to short and sharp the text.

  • The question that arises for me is whether the system can be transferred to other clinics at all? All the time, even in the title, there is talk of "in-house". So it only works in Naples? What interest would anyone else have in it? Is this just a narrative report about how it is done in Naples?

Thanks for the comment we have provided to clarify the steps of the workflow to make the procedure more reproducible in other clinics. To reproduce the procedure, the clinic in question simply needs to have the software described for medical use and a 3D printer.

  • What is the scientific question or the as yet unsolved problem behind it anyway? That is not mentioned anywhere. So: for a scientific publication you need a description of a problem that has not yet been solved. Then comes the description of the solution. Then the strengths and weaknesses of the proposed solution. And then the suggestion of how this can be transferred to other clinics.

Thanks for the suggestion, we have better clarified the clinical problem underlying the research (the lack of the occlusion as surgical guide to reduce the fractures in atrophic edentulous mandibles); we have better explained the complexities of the traditional treatment of fractures in edentulous jaws in more details and better defined the advantages of the our proposed protocol.

We hope that the revised version can be enough and more interesting for you according to the changes you have requested.

Reviewer 2 Report

The authors describe virtual planning as helpful instrument in reduction of fractures of edentulous atrophic mandibles. The article may be of interest to the readers of the journal. However, there are some minor remarks I have to made before it may be accepted for publication:

First of all; meticulous language review has to be done, avoiding grammatical and typing errors (e. g. line 301 "The successful of the treatment was confirmed")

I am missing some discussion points. Do the authors think that manually bending may weaken the plate compared to a patient specific plate?

Do the authors think that their proposed method may be helpful also for other types of mandibulart fractures, maybe comminuted fractures, in order to shorten the time of surgery?

These points should be addresses during revision of the submission.

Author Response

Dear Sir/Madam and Colleagues, thank you very much for your suggestions and queries regarding our paper. It can be remarkably improved.

  • First of all; meticulous language review has to be done, avoiding grammatical and typing errors (e. g. line 301 "The successful of the treatment wasconfirmed")

Thanks for the suggestion, we have reviewed the text with a native speaker’s help and we have eliminated grammatical errors making the manuscript more scientific.

  • I am missing some discussion points. Do the authors think that manually bending may weaken the plate compared to a patient specific plate?

Thanks for the excellent suggestion, we have provided to include this aspect within the limits of the study; the used plates were 2.4 reconstruction-plates and were modeled using the tools present in the certified systems of instruments companies. However a possible reduction in resistance could exist compared with a custom-made plate, that must not be modeled, but this reduction is less than the conventional method, because the plate is subject to less stress, given that it is modeled on a template.

  • Do the authors think that their proposed method may be helpful also for other types of mandibular fractures, maybe comminuted fractures, in order to shorten the time of surgery?

Thanks for the interesting suggestion, we have included this hypothesis in the discussion section. We are testing this protocol also for the comminuted fractures as the difficulties to manually reduce multiple fragments.

We hope that the revised version can be enough and more interesting for you according to the changes you have requested.

Round 2

Reviewer 1 Report

Can you upload a Word-File with tracked changes?

In the latest version of the manuscript, only a very few words are highlighted in red. If this was all that you have changed, the paper would be rejected.

Author Response

Dear Reviewer,

We have made revisions as usual using the word revision tool. We see 295 insertions and 270 deletions from the previous manuscript. We have re-submitted the file containing the revisions. I hope you can view it correctly and that the revisions meet your observations